# Toward an Applied Cyber Security Solution in IoT-Based Smart Grids: An Intrusion Detection System Approach

**DOI:** 10.3390/s19224952

**Published:** 2019-11-14

**Authors:** Xiao Chun Yin, Zeng Guang Liu, Lewis Nkenyereye, Bruce Ndibanje

**Affiliations:** 1Facility Horticulture Laboratory of Universities in Shandong, WeiFang University of Science & Technology, Shouguang 262700, China; xiaochunyin@wfust.edu.cn; 2College of Computer Science and Engineering, ShanDong University of Science and Technology, Qingdao 266590, China; 3Department of Computer and Information Security, Sejong University, Seoul 05006, Korea; nkenyele@sejong.ac.kr; 4Research and Development Center, Cyber Threat Intelligence Lab, YangJae Innovation Hub, 114 Taebong-Ro, Seocho-Gu, Seoul 06754-601, Korea; ndibabruce@gmail.com

**Keywords:** sensor networks, smart grid, IoT, wireless network security, DNP3 Protocol, cybersecurity

## Abstract

We present an innovative approach for a Cybersecurity Solution based on the Intrusion Detection System to detect malicious activity targeting the Distributed Network Protocol (DNP3) layers in the Supervisory Control and Data Acquisition (SCADA) systems. As Information and Communication Technology is connected to the grid, it is subjected to both physical and cyber-attacks because of the interaction between industrial control systems and the outside Internet environment using IoT technology. Often, cyber-attacks lead to multiple risks that affect infrastructure and business continuity; furthermore, in some cases, human beings are also affected. Because of the traditional peculiarities of process systems, such as insecure real-time protocols, end-to-end general-purpose ICT security mechanisms are not able to fully secure communication in SCADA systems. In this paper, we present a novel method based on the DNP3 vulnerability assessment and attack model in different layers, with feature selection using Machine Learning from parsed DNP3 protocol with additional data including malware samples. Moreover, we developed a cyber-attack algorithm that included a classification and visualization process. Finally, the results of the experimental implementation show that our proposed Cybersecurity Solution based on IDS was able to detect attacks in real time in an IoT-based Smart Grid communication environment.

## 1. Introduction

To provide new services and offer new features with excellent quality, modern critical infrastructure such as power plants, smart grids, and water plants nowadays use ICT technologies [1]. However, even if ICT technologies have made possible the provision of new services and new features, the types of connectivity have opened the door to a new wave of possible threats to critical installations. Extensive research has been performed in which the details of the vulnerability and security that affect SCADA systems have been analyzed [2,3,4,5].

In Smart Grids, protocols and architectures are designed for very particular functions in SCADA communication systems. The SCADA acronym stands for Supervisor Control and Data Acquisition, and it allows the supervision and control of plants either remotely or locally using hardware and software dedicated to that system. With this system, the system analyzes, collects and processes data in real time. The DNP3 (Distributed Network Protocol) protocol is one of the most widely used network protocols in smart grid communication networks. This protocol is mostly used in academia and industry research projects, as it provides opportunity for customization as it is an open protocol. Based on this characteristics, any company can employ DNP3 developments that are compatible with their equipment. There are other protocols designed to control the operations of technical systems. In the case of the DNP3 protocol, it is based on the three layers model in the OSI 7 layer model as given in Figure 1.

The DNP3 plays a vital role in the smart grid, and it is conventionally used in SCADA industrial processing, including both electricity and water distribution. For this reason, in this paper we focus on this protocol due to its importance in smart grids. The DNP3 protocol is based on a three-layered, enhanced performance architecture (EPA) reference model. The EPA defines the basic application functionality for the user layer, which is located between the OSI App Layer and App Program [6,7]. According to Figure 1, the DNP3 protocol mainly consists of three layers: the application layer, the data-link layer, and the pseudo-transport layer. The DNP3 protocol facilitates reliable communication between the SCADA nodes; for example, the last layer is in charge of the transmission of enormous quantities of data [8,9]. The proposed research emphasizes the DNP3 protocol by parsing its structure during the session, and we will develop an algorithm to analyze the vulnerabilities that will include modeling attacks on all layers, and we propose an intrusion detection system to detect those attacks.

Normally, a SCADA system is made up of three main layers or components such as Information Technology Network (IT Network), Operation Technology Network (OT Network) and Process Control System Network (or Field Layer), as given in Figure 2.

In recent years, the OT Networks were operated as separate networks or stand-alone system without being connected to public communication and infrastructures. Nevertheless, as the Internet nowadays provides data accessibility and connected services, the businesses in Smart Grid have turned to exploit those services. This situation creates a complex architecture where non-secure systems are added to existing systems without strong security. Ultimately, they are both more exposed to attackers, as the separation that had previously protected these systems is decreased [10].

Unfortunately, with the presence of internet connections inside smart grids, the security risk is very high. After what happened in Ukraine, everyone knows that hackers can bring down the energy grid. Attacks often happen by using steps such as reconnaissance, which consists of gathering information about the targeted system; scanning, which is about finding any weakness or vulnerability in the system by looking for any open ports; and running a service through the port. Thirdly, the attacker exploits the system using the discovered vulnerability and then compromises it so that they can gain full control. Finally, the attacker tries to maintain access by which they will steal the data or damage the system whenever they want [11,12]. These types of attacks are not easy to detect using traditional antivirus software, which detects malware using pattern matching or heuristic methods for obfuscated malware. Cyber-attacks have evolved in business-driven situations, because the criminal actors behind these attacks know where the system weaknesses reside, and they employ appropriate malware, especially ransomware.

As previous experience shows, whenever there is a hack on a smart grid, people’s life is in danger, and sometimes loss of life occurs.

To tackle malware threats on the SCADA system, especially on the smart grid, this article presents an intrusion detection system that isn’t based on signatures, like traditional antivirus software. The proposed solution is mainly based on the DNP3 protocol, where the engine parses the packet format and then we train it to learn whether the sample from the frame protocol has been compromised or is good. The advantage of this protocol is that it is used in oil/gas and water utilities, as well as in wastewater. Furthermore, it is broadly utilized in electric facilities such as catapults [13]. The research contribution in this article includes the main steps as follows:An overall customized DNP3 protocol vulnerability exposure with reference to the original protocol.A new attack model using the DNP3 protocol that targets all layers; the attack follows three steps: (a) pre-attack step; (b) attack modeling defined on all layers; and (c) attack settings within DNP3 parameters and consequences.An algorithm that includes machine learning methods for data transformation and data process concept on SCADA/DNP3 protocol.A new cyber-attack algorithm targeting the SCADA/DNP3 system, and the visualization and classification process for an intrusion detection system (IDS).

In this paper, Section 2 presents a description of existing solutions in the area, while Section 3 presents the proposed solution along with the methods and data collection in an experimental environment. The results of experimental testing and the discussion are provided in Section 4, before concluding the work in Section 5.

## 2. Related Work

This section describes the works—including various solutions applied in SCADA/ICS/DNP3—in which IDS and other methods have been discussed. The Distributed Network Protocol (DNP3) is prevalent within critical infrastructure, especially in smart grids. Unfortunately, DNP3 has some vulnerabilities that have been exploited by hackers, and so SCADA systems would face serious problems [14]. However, before exploring solutions in SCADA systems, there is a great deal of research and many excellent resdults in IoT networks that are very promising. Yang et al. [15] proposed a security scheme in IoT-based healthcare systems. In this research, they proposed a self-adaptive access control together with a privacy-preserving smart IoT-based healthcare big data storage system. Further security approaches have been developed for IoT systems, as described in [16,17,18].

With regard to machine learning methods, de Toledo et al. [19] developed a method that encrypts the traffic using the DNP3 protocol. This study used supervised algorithms to classify messages from the same protocol using datasets from the medium voltage of substations using simulation methods. The generated traffic followed two-direction communication using an encryption mode based on the IPsec and ESP (transport mode) with the exclusion UDP mode. This experiment is the most widely used, as it provides a way of privately limiting the cost of IP bandwidth within networks per byte sent [20]. Other techniques that have been proposed as solutions for protecting DNP3 traffic include statistical pattern recognition, classification-based real-time method with HTTP, FTP and SSH flow, TCP and TLS protocols [21,22].

Widely known layered security methods that provide protection in SCADA networks have been developed, but these methods have numerous limitations with respect to their dependency on the protocols. Among others, protocols such as SSH, SSL, IPSec, and TLS offer end-to-end security solutions, in addition to crypto-protocol encryption systems [23,24]. Further research oriented towards the security of the application layer [25] focusing on data integrity and authentication procedures was developed with the aim of providing solutions for known attacks such as modification, spoofing, and flooding [26]. Nevertheless, a certain number of limitations was revealed resulting from mechanisms defined in the DNP3 protocol—in particular, embedded security mechanisms [27]. A solution based on crypto-algorithms that includes known encryption methods like AES and RSA was developed to protect DNP3 protocol at the application layer [28]. In this research, the authors contributed three primary enhancements, including a new security scheme that was implemented together with the DNP3 protocol, a method for constructing the bytes within every layer, and use of the TCP/IP protocol for data exchange.

On the other hand, IDS based on different machine learning methods has been developed, whereby attacks can be detected based on highly accurate results of detected attacks. However, more improvements are necessary due to false alarms or false positive from the detection systems. This problem usually leads to the misclassification between good and bad data in the network [29]. In the same category of research, a group of five machine learning algorithms was tested for cybersecurity solutions to protect SCADA systems [30,31,32,33]. After the training process, the models were implemented in a real network environment to capture and analyze online data from network traffic. Both results from the testbed and live traffic revealed that the IDS based on machine learning algorithms was efficient for detecting attacks. Further research developed by Keliris et al. [34] showed that the Support Vector Machine (SVM) algorithm performs well for anomaly detection and classification. They used a supervised learning method to develop a process-aware defense tactic in the ICS accounting for behavior-based attacks. The work done in [35] suggests that a detection system using machine learning techniques in power systems would be feasible for detecting malicious states. Tomin et al. [35] claimed that such techniques, where applied in SCADA/ICS, offer a range of solutions with a satisfactory level of security. In the course of their research, they used an offline training process using a cross-validation method and they applied it to a semi-automated method for online testing purposes. Further research has been developed to provide security for Smart Grid DNP3, through the identification of malicious activities in ICS of IoT based on Deep Learning, IDS for SCDA systems, and Neural Network-based IDS for critical infrastructure. These have shown tremendous results in the development of models for the detection of attacks on power systems [36,37,38,39,40,41].

## 3. Proposed Solution: Method and Implementation Experiments

This section describes the proposed scheme for the SCADA/DNP3 protocol. The solution requires several steps, referred to as “modules”, and each of these plays a specific role in building a holistic cyber-security solution in an IoT-based Smart Grid environment.

### 3.1. System Model and Description

The proposed solution is based on the following modules: (a) data input system, (b) data analysis system, and (c) classification and detection system, as shown in Figure 3. However, before we could arrive at this holistic solution, we performed additional research on the DNP3 protocol. Firstly, we developed an attack model for each layer of the DNP3 protocol, as shown in Figure 4. These attacks had two mains functions: (1) to collect data for the purpose of building a database to be used in the training and testing model, (2) to assess the vulnerabilities of the DNP3 protocol [42] that attackers are able to leverage in order to carry out cyber-attacks on IoT-based Smart Grids. Secondly, we developed an algorithm for analyzing a modified DNP3 protocol [43,44]. This algorithm uses the original DNP3 protocol as a reference for the purpose of comparison with the common vulnerabilities of the protocol stack.

We used four types of attack—modification, interception, interruption, and fabrication—targeting all layers in order to evaluate them. The collected vulnerabilities (based on the attacks on the two protocols) were used with a mapping function to modify the features of the DNP3 protocol. The results provide the vulnerabilities discovered for the customized protocol, as shown in Figure 5.

### 3.2. Data Input System: Data Generation

The dataset used in the experiment is from a variety of different sources, but the most important data, which is related to DNP3 packet parsing, was based on the assessment of vulnerabilities and attacks performed on the protocol. Therefore, we will only describe the data obtained from these experiments, as this is the focus of our research. The rest of the data was obtained from an open-source dataset used at the 4SICS industrial cybersecurity conference [3], which is an annual summit. The summit calls on experts in ICS/SCADA/DNP3 cybersecurity from the most critical infrastructures like smart grids, transportation, and so on. As far as our purposes are concerned, we only collected data—PCAP Files—related to the smart grid from the ICS Lab using RTUs, PLCs, and other industrial networks. Additionally, we included some known malware targeting ICS/SCADA systems. As described above, to generate the data, several steps are required, which can be summarized as two main steps: vulnerability assessment and attack modeling on the DNP3 protocol.

#### 3.2.1. DNP3 Protocol Vulnerability Assessment

Different methods have been proposed in order to analyze the weaknesses in the DNP3 protocol; one of these methods presents an assessment of specific attacks on function code within certain layers of the protocol stack [45,46,47]. In this paper, we used a customized DNP3 protocol to perform the vulnerability analysis, and this is compared with the original one, as shown in Figure 4. The novelty of our method is that we map common vulnerabilities onto the customized database features, with the results showing us the weakness of the protocol, meaning that we can ultimately launch different attacks in order to collect or generate the data to use in our experiments.

The proposed algorithm takes the two protocols as input and checks whether they satisfy the protocol stack requirements. If they do, they are parsed into the main layers; otherwise, they go back to the starting point. This process continues, using DNP3 threats such as modification, interruption, interception, and fabrication, where we define the common vulnerabilities of the protocol stack to to be used for the mapping process. The mapping process is carried out based on a database of features from the DNP3 packet. Table 1 shows selected features from the layers of the DNP3 protocol, and a full account of the features is provided in Appendix A. A “Yes” in the column “Subject to Attack” means that they present a potential weakness that makes them vulnerable to DNP3 protocol threats.

#### 3.2.2. Attack Modeling on the DNP3 Protocol

To launch attacks on the DNP3 protocol, we made an attack model that was specific to the vulnerabilities discovered. As shown in Figure 5, the model is based on three main steps:✓Step 1: Pre-attack. This is where the preliminary is carried out, including obtaining the DNP3 packets from the repository, and preprocessing the packet in order to obtain three layers for the next step.✓Step 2: Attack Modelling. In this step, we define the attacks on the basis of the vulnerabilities discovered in each of the following layers: Data Link Layer, Transportation Layer, and Application Link Layer. For the first layer, we defined three attacks (Length Overflow Attack, DFC Flag Attack and Reset Function Attack), for the second layer, we defined two attacks (Fragmented Message Interruption and Transport Sequence Modification), and in the last layer, we defined two attacks (FC Modification Request and Configuration Capture Attack IIN).✓Step 3: Setting up the attack with DNP3 parameters and Consequences. This step defines the parameters to be used during the attack (payload) and describes the consequences of each attack.

As given in the description of the consequences, each attack leads to bad behavior in the smart grid network. The aim is not to have these attacks, but rather to develop countermeasures in order to protect the network, devices, data, and human beings. Both the vulnerabilities and the attacks have several operational impacts that could cause damage to the system or take over the control system [48,49,50,51,52].

The data input system consists of malware and benign data, as already described in the introductory paragraph of this subsection. Table 2 gives a summary of the dataset used in this paper, where the name column describes the name of the malware or benign data, Qt is the amount of each type and the percentage of the distribution over the total. The overall distribution of malware is 55%, and that of benign data is 45%, which is acceptable for a classification and detection model. Bencsath et al. [53] described the most dangerous malware targeting industrial infrastructure in detail. Stuxnet was discovered in 2010, when it was reported to have destroyed numerous centrifuges in Natanz. The centrifuges had been designed for a uranium enrichment facility in Iran. The infection vector of Stuxnet was the USB, from which the worm was installed on and spread among interconnected computers. It is therefore very important to produce a cyber-security solution based on the IDS and ML techniques in order to protect such critical infrastructures against malware.

### 3.3. Data Analysis System

This module is located in the middle of the other modules, as it takes the input from the various repositories and then transforms the data into a format compatible with the functions of the next module. The data analysis is built up over many steps and requires advanced knowledge of Data Science, with several tools to be used in such work. In this paper, we describe a few steps taken from Figure 6. The data analysis consists of eight steps, from raw data input to the visualization step.
Step 1: This is the initial action, where module one feeds raw data to the second module. As described above, 55% of the dataset is made up of malware and 45% is made up of benign data.Steps 2 to 4: After getting the raw data, the engine proceeds to DNP3 protocol extraction with the integration of various fields with pre-processing actions such as contextualization and mapping in order to prepare for loading to the DB. Before that, the engine carries out the data cleaning, removing unwanted fields, carrying out de-noising, and nullifying some fields that match with the DB used in our experiment.Steps 5 to 8: This is where the engine utilizes the DB constructed in Steps 2 through 4. At this stage, the important features are extracted based on their presence in the DB (presence refers to how frequently this feature occurred throughout the whole DB). Because the DB is a mixture of many types of data, the classification process first requires that the data be transformed from categorical and numerical data to a binary data format. Once we have one type of data, it is possible to apply the ML algorithms directly (green arrows) and then execute the classification process.

Different algorithms were used for this process of data transformation in order to obtain the final DB and visualization results. Algorithm 1 is the pseudocode where all main steps are called to execute the ML algorithms. As given by the algorithm, we listed from Step 11 to 14 some of the algorithms used for the data transformation and visualization process. As input, we used a mixture of malware and benign data from module one, but for the paper objectives, we are going to focus only on the DNP3 packet analysis for more details. In the case of malware, we will describe Stuxnet and the features selected from the data analysis process.

Algorithm 1 has two main parts: the data input and processing part. The second part of the algorithm gives the main steps that implement the data transformation until the visualization step. The algorithm instructs to select all features from the raw data that include DNP3 protocol features, the 5 tuples (Source IP, Destination IP, Source Port, Destination Port, and Protocol), and eventually the features of the malware raw data. Next, it does a format check, which requires removing some unwanted characters that would cause errors in the database. In this case, null fields are not allowed, and categorical data and numerical data have to be mapped too. The cleaned data will then constitute the initial database, where we can make some queries to see the content.

The biggest part of the algorithm is where the call of each machine learning algorithm is running for different functions. Feature selection is the most important step in malware classification when using an ML algorithm. Bugra et al. [54] presented a method for malware classification where they applied DL (Deep Learning) methods. The authors performed the classification of malware based on a shallow deep learning network. To realize their experiment, they used a two-layer neural net to process the text, which consisted of turning text into a numerical form that is understandable by deep networks. This is called word2vec, developed by Tomas Mikolov [55,56] at Google and which is available from the Google code archive [57].

The work in [58] gives methods where ML has been used to classify malware and detection, in addition to implementations directions. The main goal of their work is to give a list of best classification methods such as feature selection, representation using Cuckoo Sandbox, k-Nearest-Neighbors (KNN), Decision Tree (DT), Support Vector Machines (SVM), Naive Bayes and Random Forest.


**Algorithm 1 Data Transformation & Visualization**


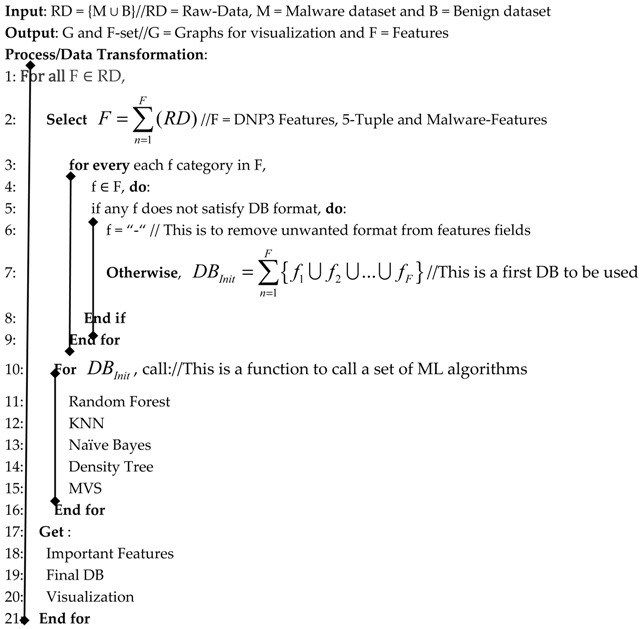



In this paper, we have used many algorithms, such as k-Nearest-Neighbors (KNN), Decision Tree (DT), Support Vector Machines (SVM), Naive Bayes and Random Forest. The results from our experiments and their descriptions are presented in Section 4.

### 3.4. Cyber-Attack Algorithm and IDS Solution

This subsection describes the cyber-attack algorithm that we created, in addition to the countermeasure (the IDS to detect the attack). This final step leads to the classification and detection processes from Figure 3. After the vulnerability assessment of the DNP3 protocol, the attack modeling, and data collection, we have now all we need to launch the attack and then perform the classification and detection solution. Algorithm 2 gives the steps to launch an attack on DNP3 protocol.


**Algorithm 2 Cyber-Attack on DL TL and AL**

**** START ****
01: **Input** ← Raw data02: **Output** ← Anomaly and Normal Traffic: {Classification and Detection}
**** PRE-ATTACKS ****
03: Procedure: INTERCEPTION (I)04: Action: INJECTION (Inj) or MODIFICATION (Mod)05: Packet ← {pre-process, get DNP3 packet *(dnp_pkt_)*}06: DNP3 protocol ← {DataLink (DL), TransportLink (TL), ApplicationLink (AL)}07: Attack = {LOVA,DFC,FCA,FMI1,FMI2,TSM,FCM,CC_IIN}
***** SETTING-UP PARAMETERS & ATTACK LAUNCHING *****
08: LOVA ⟺ $I_{\left(dnp_{pkt}\right)}\left\{Mod\left(DL_{length}\leftarrow DL_{length\pm \alpha }\right)\right\}$09: DFC Flag ⟺ $I_{\left(dnp_{pkt}\right)}\left\{Mod\left(DL_{DFC=0}\leftarrow DL_{DFC=1}\right)\right\}$10: FCA ⟺ $I_{\left({dnp}_{pkt}\right)}\left\{Mod\left(DL_{FC}\leftarrow DL_{FC=1}\right)\right\}$11: FMI1 ⟺ $I_{\left({dnp}_{pkt}\right)}\left\{Inj\left(TL_{FMI(FIN)}\leftarrow TL_{FMI(FIN=1)}\right)\right\}$12: FMI2 ⟺ $I_{\left({dnp}_{pkt}\right)}\left\{Inj\left(TL_{FMI(FIR)}\leftarrow TL_{FMI(FIR=0)}\right)\right\}$13: TSM ⟺ $I_{\left({dnp}_{pkt}\right)}\left\{Mod\left(TL_{TSM(Seq.Number)}\leftarrow TL_{TSM(Seq.Number\pm \beta )}\right)\right\}$14: FCM1 ⟺ $I_{\left({dnp}_{pkt}\right)}\left\{Mod\left(AL_{FC}\leftarrow AL_{FCM.req=0x02}\right)\right\}$15: FCM2 ⟺ $I_{\left({dnp}_{pkt}\right)}\left\{Mod\left(AL_{FC}\leftarrow AL_{FCM.req=0x0d}\right)\right\}$16: FCM3 ⟺ $I_{\left({dnp}_{pkt}\right)}\left\{Mod\left(AL_{FC}\leftarrow AL_{FCM.req=0x0e}\right)\right\}$17: FCM4 ⟺ $I_{\left({dnp}_{pkt}\right)}\left\{Mod\left(AL_{FC}\leftarrow AL_{FCM.req=0x14}\right)\right\}$18: FCM5 ⟺ $I_{\left({dnp}_{pkt}\right)}\left\{Mod\left(AL_{FC}\leftarrow AL_{FCM.req=0x15}\right)\right\}$19: FCM6 ⟺ $I_{\left({dnp}_{pkt}\right)}\left\{Mod\left(AL_{FC}\leftarrow AL_{FCM.req=0x82}\right)\right\}$20: CC_INN ⟺ $I_{\left({dnp}_{pkt}\right)}\left\{Mod\left(AL_{CC\_IIN}\leftarrow AL_{CC\_IIN=1\to 5^{th}bit\;in\;2^{nd}byte}\right)\right\}$21: **If**
$I_{\left({dnp}_{pkt}\right)}$ then, //Interception of the packet22: Attacker $I_{\left({dnp}_{pkt}\right)}\left\{{\left(Mod∥Inj\right)}_{DL\;\neg TL\;\neg AL}\right\}$ //Launch the attack on the layers23: **Get** anomaly traffic24: **End** if25: **End**

All of these attacks on DNP3 protocols are assumed to occur during data transmission from one station to another. In practical cases, the system uses Master (client) and Slave (server) terminology. In this case, the server is defined as a station or device that holds and processes the information needed by an operator. To the other side, a client is a substation or device that requests information from the server. The DNP3 protocol provides the ability to facilitate data transmission between Master and Slave [59]. Figure 7 shows a schematic attack in which the attacker performs interception, modification and injection attacks on the DNP3 packet content as described in Algorithm 2.

The abovementioned algorithm was built up over three stages: start, pre-attack and setting parameters up & attack launching. The last stage includes seven major attacks that target all three layers on specific fields. The second stage instructs the algorithm to use the interception method that executes the injection and modification attacks in addition to the three layers and corresponding attacks. The first stage is related to data input and output data format information. The abbreviations used in the algorithm are described in Table 3.

The main part (stage number 3) of the algorithm describes the attacks that target the layers as follows:From Steps 8 to 10: These are the sets of attacks aiming for the modification of the Data Link Layer parameters by intercepting the DNP3 packet. The actions that are carried out as part of the Length Overflow Attack (LOVA), Data Flow Control Flag (DFC Flag) and Function Code Attack (FCA) are executed using the interception procedure. During the attack, the length is modified by ± α to the original size, the DCF and FCA are modified with 0 or 1.From Steps 11 to 13: The targeted layer is the Transport Link Layer with the injection action of some fault parameters to the DNP3 packet. At this stage, the Fragmented Message Interruption Attack (FMI1 and FMI2) supports the fault parameters by injecting 1 or 0 to the First (FIR) and Final (FIN) Bit Number. The Transport Sequence Modification Attack (TSM) is also one of the TL attacks with sequence modification by ± α to the original order, but it is based on the modification procedure.From Steps 14 to 20: This is a range of attacks on the Application Link Layer with a large number of parameters. After the DNP3 is intercepted, the modification process is performed on the packet at the Application Layer. To do so, the Function Code Modification Attack (FCM1~6) is called, where the attacker sets up the parameters to be modified. The request to modify this function code at the application layer is based on the selected values (such as 0x02, 0x0d, …, 0x82 and modification of a byte of the internal indication, such as the 5th bit in the 2nd byte of the DNP3 packet at the Application Layer).

After the last step, the whole DPN3 packet (in the current session) is compromised, and it is time that the engine can classify between bad and good traffic. The results from the experiment are detailed in Section 4.

## 4. Experimental Results and Discussion

### 4.1. Malware Sample Feature Selection Results

The following section discusses the findings after data transformation for the classification process. For the purposes of our paper, we cannot include all of the figures and tables, but we have selected the most important results from among others. As described above in Section 3, the input data comprised about 10 malware and 80 of the benign dataset, which represents 56% malware and 45% benign data, respectively. For malware analysis and feature extraction, we selected the Stuxnet malware, and we parsed this sample using the Pepper tool, which is an open-source tool for malware static analysis on a portable executable [60]. We extracted the metadata, header, opt header, sections, and import features from the executable file, as shown in Figure 8.

The Stuxnet malware PE result shows that many system files are subject to compromises or attacks. Figure 9 shows the distribution of the Top 20 process names found after the reverse engineering of the malware using the Pepper tool. We selected only major information, with 50% and high score points as given in Table 4. The main reason is that after computing all of the features, it was necessary to statistically pick out only those with a high degree of presence in the original database. Presence refers to how frequently the feature occurs throughout the whole DB. For our experiment we set, 50% as the threshold. The table indicates that the malware target memory process has highest score, with four times, and it can be observed that the file names being compromised are related to the memory processes. The other process is related to the local security system authority service, which is a highly critical system file in Microsoft Windows (the lsass.exe). Most malware targets this file, because it is used to enforce security policies related to sensitive information such as password changes and login access verifications. The malware also targets another executable file with a task of high importance in the Windows Task Manager, and which contains machine code, and this is called vdmdbg. It has also a high score in the below table. Appendix B provides all of the feature information from the Stuxnet Portable Executable (PE) file.

### 4.2. DNP3 Protocol Packet Sample Feature Selection Results

The DNP3 packets that include the attack types defined in Section 3 are collected using the Wireshark tool, which is a network packet analyzer that captures network packets and displays the packet contents with the maximum detail possible [61]. In order to generate the packet, we developed an exploit that was specifically designed to carry out a cyber-attack on the DNP3 protocol. This malicious software is real, and we advise the reader of this paper not to try this on a live product. The exploit, as given in Appendix C, carries the data (payload) that intercept the traffic and then injects some modified parameters, as described in Algorithm 2.

After the attacks, we collected the features from the DNP3 packet where the results revealed that the predicted attacks (as defined in Algorithm 2) achieved the goals. Table 5 describes our experimental results in detail, along with the impact on the SCADA/DNP3 devices. As can be seen, the impact depends on the attack type, the parameter modified in the original format, and the link layer that is attacked. As described above, it is prohibited to run the provided exploit in a real working environment, because the impact of the attack would be damaging. The rest of the features of DNP3 are given in Appendix A.

### 4.3. Visualization and Classification

The discussion in this subsection is related to the results of the proposed methods based on the classification of malware, which is displayed in the form of a graph visualization. Table 6, with Figure 10 and Figure 11, describes the classification results with the following explanation:Login-Time: The field indicates when the event happened. It is, therefore, easy to track down and find out the right moment for the attack on the system when there is a need for an investigation.Source-IP: Every traffic on the network includes the source IP address, which indicates the origin of the data, request, or other type of transaction. In our experiment, this is the IP address of the device that is sending information to the destination device.Source-Port: This is one of the user session parameters that tells the system where to reply to the response. It is always associated with the source-IP and the different applications and protocols used by the sender.Destination-IP: This is where to go. The receiving device in our experiment has a destination IP to which the packet is to be sent. This enables two-way communication in the configuration.Destination-Port: The same explanation as the source port, except that this is for the destination device.Classification: As stated before, the aim is to distinguish between benign and malware groups for elements in the dataset. Hence, after the process, the result results in an “anomaly”, as bad packets related to malware or any malicious activity are discovered during the analysis. We only provide those results that identify an anomaly.Field: With this information, we can see what type of feature, attack type or any other field has been targeted. In this case, the system gives “Transport FIR”, which indicates the DNP3 protocol feature.Graph Visualization: The 3D graph indicates the classification as either malware or benign data. The red dots indicate the malware sample in our experiment, while the blue ones indicate the benign dataset. Additionally, there other two graphs, which give an overview of the Top 5 source and destination IP addresses.

## 5. Conclusions

This paper discussed cybersecurity solutions based on the Intrusion Detection System in the IoT-based Smart Grid. We described in detail the concept of a system based on the IoT for Smart Grids using the SCADA/DNP3 communication protocol. To achieve the proposed method, we developed and presented a series of algorithms for implementation along with experiments.

In this paper, we developed a new method for assessing DNP3 protocol vulnerability, which gave us an idea of where to perform the attack. This assessment was conducted on a modified DNP3 protocol with reference to the original protocol. Next, based on the discovered vulnerabilities, we developed the new attack model aiming at the Data Link Layer, Transport Link Layer and Application Link Layer of the DNP3 protocol. Moreover, we developed two algorithms that helped us perform data transformation using Machine Learning methods. The other algorithm includes all of the steps for the cyber-attack on the DNP3 protocol; this also includes the classification process. Finally, we presented the experimental results, showing that the proposed method was able to detect intrusions to the SCADA system based on an IoT Smart Grid and could classify them with detailed information about the compromised fields from the DNP3 packet.

## Figures and Tables

**Figure 1 sensors-19-04952-f001:**
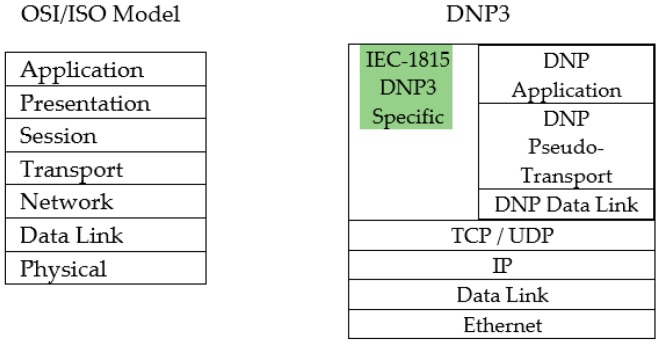
DNP3 protocol main concept with OSI/ISO mapping.

**Figure 2 sensors-19-04952-f002:**
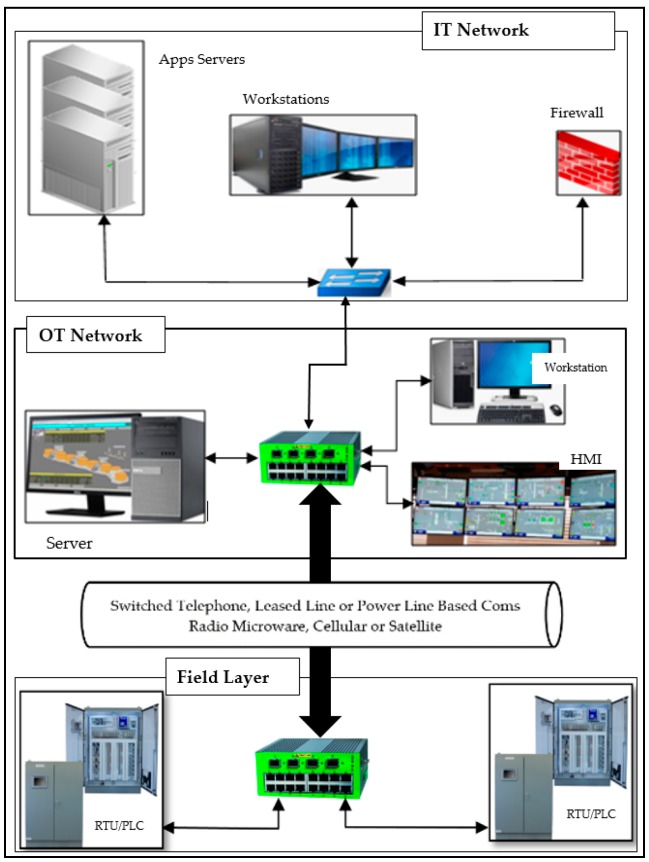
Basic SCADA concept components or layers.

**Figure 3 sensors-19-04952-f003:**
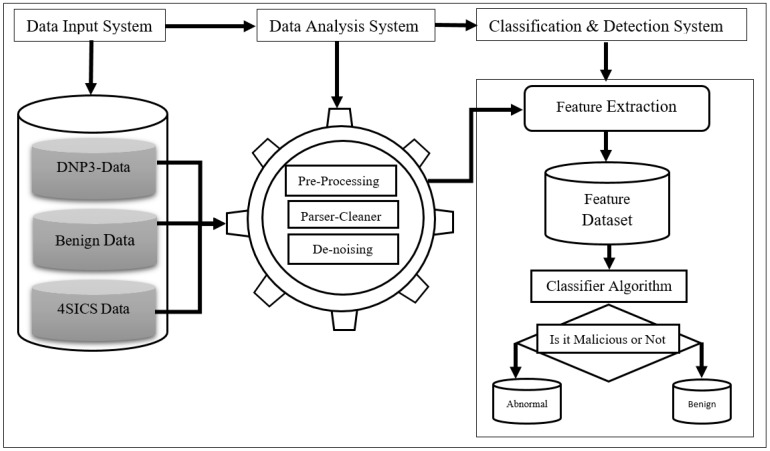
System model for the proposed solution.

**Figure 4 sensors-19-04952-f004:**
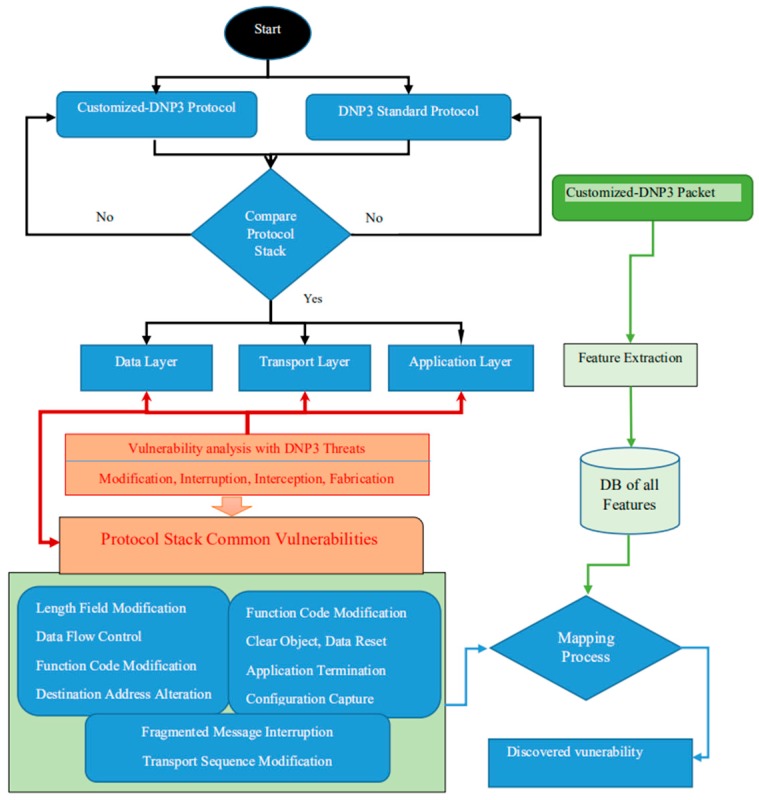
DNP3 overall vulnerability algorithm analysis.

**Figure 5 sensors-19-04952-f005:**
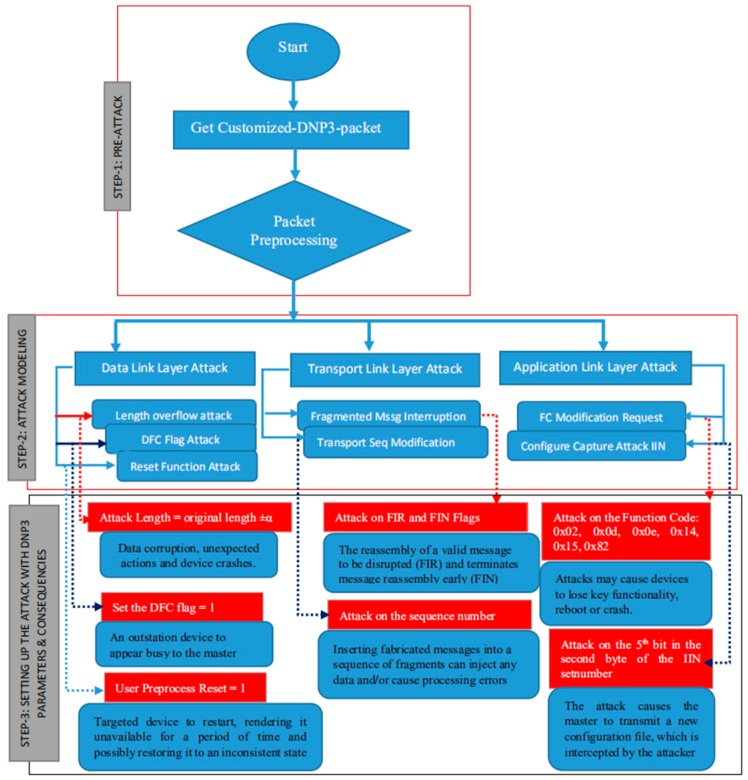
Attack modelling for the DNP3 protocol.

**Figure 6 sensors-19-04952-f006:**
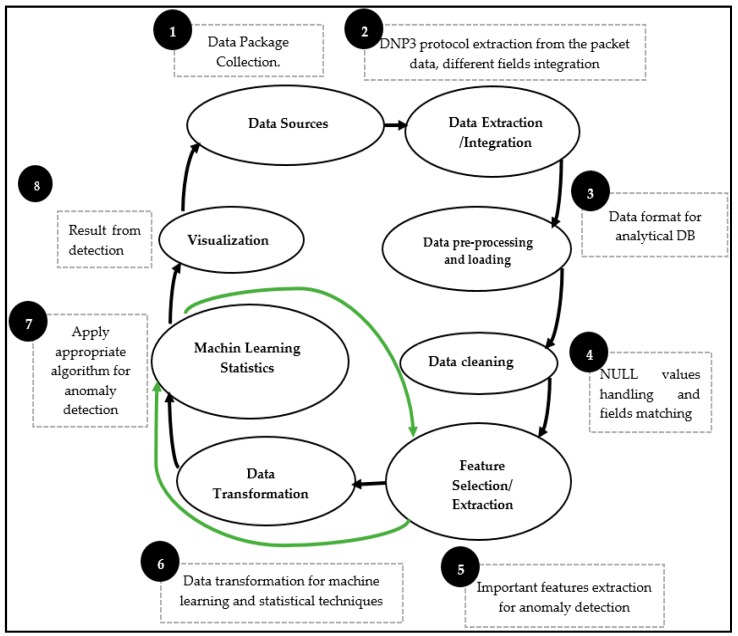
Data transformation processes.

**Figure 7 sensors-19-04952-f007:**
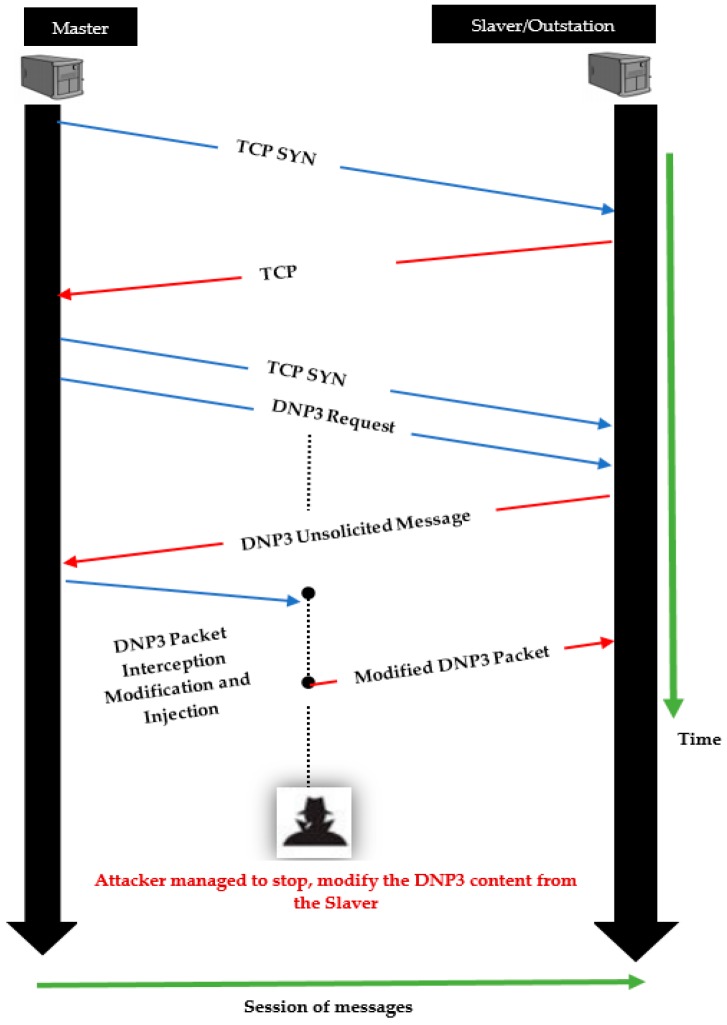
DNP3 cyber-attack types: interception, modification, and injection.

**Figure 8 sensors-19-04952-f008:**
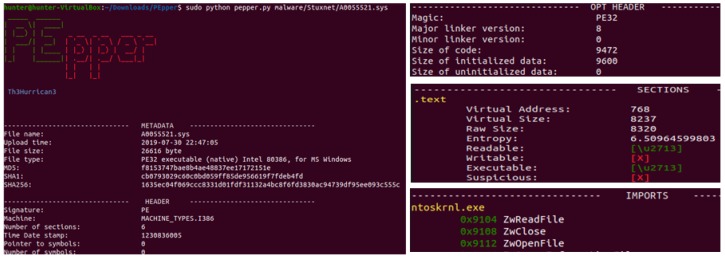
Pepper tool: Stuxnet PE malware reverse engineering and feature extraction.

**Figure 9 sensors-19-04952-f009:**
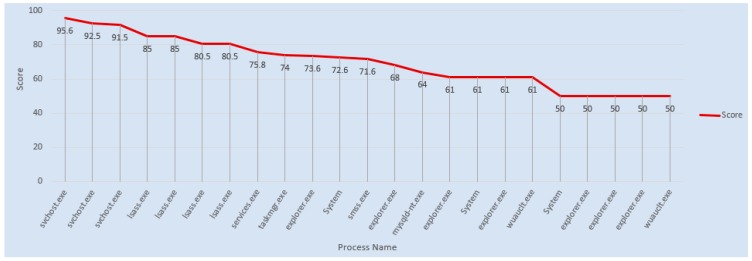
Top 20 targeted processes from the Stuxnet PE file.

**Figure 10 sensors-19-04952-f010:**
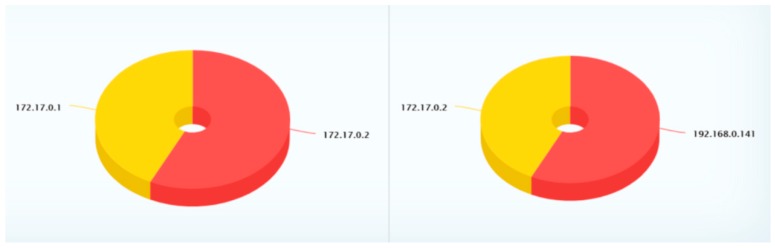
Top 5 Source IPs (**right**) and Top 5 Destination IPs (**left**).

**Figure 11 sensors-19-04952-f011:**
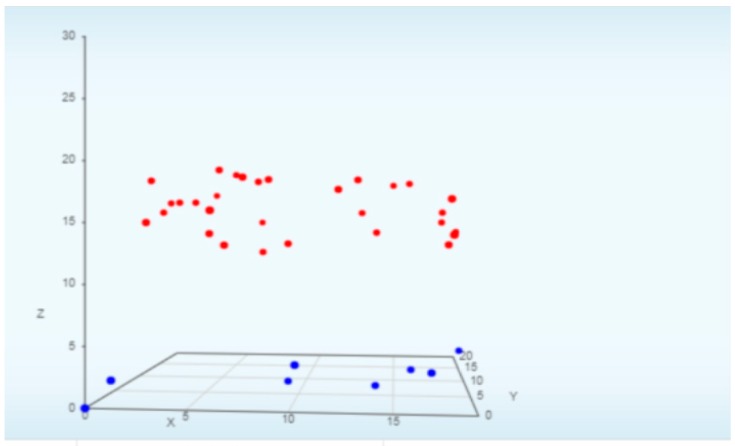
Classification with a 3D graph. Red dots are malware. Blue dots are benign.

**Table 1 sensors-19-04952-t001:** DNP3 partial features dataset.

Features	Subject to Attack
**DNP3_START**	
**DNP3_LENGTH**	Yes
**DNP3_SOURCE**	
**DNP3_DESTINATION**	
**DNP3_CONTROL_DFC**	Yes
**DNP3_CONTROL_DIR**	
**DNP3_CONTROL_FCB**	Yes
**DNP3_CONTROL_FCV**	Yes
**DNP3_CONTROL_FUNC_CODE_PRI**	Yes
**DNP3_CONTROL_FUNC_CODE_SEC**	
**DNP3_CONTROL_PRM**	
**DNP3_CONTROL_reserved**	
**DNP3_CRC**	
**DNP3_Transport_FIN**	Yes
**DNP3_Transport_FIR**	Yes
**DNP3_Transport_SEQUENCE**	Yes
**DNP3_Application_request_Application_control_CON**	
**DNP3_Application_request_Application_control_UNS**	
**DNP3_Application_request_FUNC_CODE**	Yes
**DNP3_Application_response_Application_control_CON**	
**DNP3_Application_response_IIN_CLASS_3_EVENTS**	
**DNP3_Application_response_IIN_CONFIG_CORRUPT**	Yes
**DNP3_Application_response_IIN_DEVICE_RESTART**	

**Table 2 sensors-19-04952-t002:** Dataset of malware and benign data in this paper.

Name	Malware	Qt.	%	Benign	Qt.	%
**Triton**	Yes	1650	15.46	No	0	0.00
**Industroyer**	Yes	1521	14.25	No	0	0.00
**BlackEnergy**	Yes	650	6.09	No	0	0.00
**Stuxnet**	Yes	1120	10.50	No	0	0.00
**Duqu**	Yes	944	8.85	No	0	0.00
**Flame**	Yes	1062	9.95	No	0	0.00
**Gauss**	Yes	1267	11.87	No	0	0.00
**DNP3-Data(Original)**	No	0	0.00	Yes	4593	53.58
**DNP3-Packet (Experiment)**	Yes	2456	23.02	N o	0	0.00
**4SICS**	No	0	0.00	Yes	3980	46.42
Total		10,670	55		8573	45

**Table 3 sensors-19-04952-t003:** Notations and description used in Algorithm 2.

Abbreviations	Description
**DLL**	Data Link Layer
**TLL**	Transport Link Layer
**ALL**	Application Link Layer
**LOVA**	Length Overflow Attack
**DFCA**	Data Flow Control Attack
**FCA**	Function Control Attack
**FMIA**	Fragmented Message Interruption Attack
**TSMA**	Transport Sequence Modification Attack
**FCM**	Function Code Modification
**CCA_IIN**	Configuration Capture Attack_Internal INdication

**Table 4 sensors-19-04952-t004:** Stuxnet features and scores.

Number	File Name	Process Name	Score %
1	memory-mod-pe-0 × 20000000-0 × 10124000	service.exe	95.6
2	kerner32.dll.aslr.0013a1e	svchost.exe	92.5
3	kerner32.dll.aslr.0013b86	svchost.exe	91.5
4	Memory mod-pe-0 × 00090000-0 × 0010a000	lsass.exe	85
5	Memory mod-pe-0 × 00090000-0 × 0010a000	lsass.exe	85
6	lsass.exe	lsass.exe	80.5
7	lsass.exe	lsass.exe	80.5
8	memorymod-0 × 006b0000-0 × 006b1000	services.exe	75.8
9	vdmdbg.dll	taskmgr.exe	74
10	izarccm.dll	explorer.exe	73.6
11	ntosknl.exe	System	72.6
12	ntdll.dll	smss.exe	71.6
13	olepro32.dll	explorer.exe	68
14	mysqld-nt.exe	mysqld-nt.exe	64
15	mlang.dll	explorer.exe	61
16	bhomanger.dll	System	61
17	hal.dll	explorer.exe	61
18	wuaucpl.cpl	wuauclt.exe	61
19	mrxnet.sys	System	50
20	vmhgfs.dll	explorer.exe	50
21	odbc32.dll	explorer.exe	50
22	wuaucpl.cpl	explorer.exe	50
23	odbc32.dll	wuauclt.exe	50

**Table 5 sensors-19-04952-t005:** Feature description of DNP3 protocol attack.

DNP3-Features	Description	Attack Type	Parameter	Impact	Link Layer
**DNP3_LENGTH**	Length of field	LOVA	Original length modification	Device crashes	DL
**DNP3_CONTROL_DFC**	The DFC tells other devices that the current device is busy	DFC	Flag = 1	Eternal busy	DL
**DNP3_CONTROL_FUN_CODE_PRI**	Primary Function code	User Process reset	Code = 1	Unwanted restart	DL
**DNP3_Transport_FIN**	Final bit	FMI	Flag modification	Early message termination	TL
**DNP3_Transport_FR**	First bit	FMI	Flag modification	Message processing error	TL
**DNP3_Application_request_FUN_CODE**	Function Code	FCA	0 × 02, 0 × 0d, 0 × 0e, 0 × 14, 0 × 15, 0 × 82	Crash or reboot	AL
**DNP3-Application_response_IIN_CONFIG_CORRUPT**	Configuration File System	CC_IIN	5th bit in 2^nd^ byte	Configuration file modified	AL

**Table 6 sensors-19-04952-t006:** Classification result description.

Source-IP	Source-Port	Destination-IP	Destination-Port	Classification	Field
**172.17.0.1**	59686	172.17.0.2	45000	Anomaly	Transport FIR
**172.17.0.1**	59686	172.17.0.2	45000	Anomaly	Transport FIR
**172.17.0.2**	41044	192.168.0.141	45000	Anomaly	Transport FIR
**172.17.0.2**	41044	192.168.0.141	45000	Anomaly	Transport FIR
**172.17.0.2**	41044	192.168.0.141	45000	Anomaly	Transport FIR
**172.17.0.2**	41044	192.168.0.141	45000	Anomaly	Transport FIR
**172.17.0.2**	41044	192.168.0.141	45000	Anomaly	Transport FIR

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
