# Peer review of "Toward an Applied Cyber Security Solution in IoT-Based Smart Grids: An Intrusion Detection System Approach"

_sensors, 2019, doi:10.3390/s19224952_

Round 1

Reviewer 1 Report

It seems that this paper is a technical report on cybersecurity solutions based on intrusion detection system in smart grid.

This paper should thoroughly be rewritten and reorganized. Throughout this paper, it is very hard for the reviewer to get the main idea that the authors want to describe.

Moreover, this paper lacks of theoretical depth.

Reference 21 is missing.

It is not suitable for publication in Sensors.

Author Response

Dear Valued Reviewer,

Thank you for your time to review our article. Without your noble work, our article could not move forward to this stage. We really appreciate your excellent work.

Hereby, we would like to transfer our answers to your questions. Once again, with your questions, the article has been updated accordingly. Should you find any other recommendation or suggestion, please let us know, we will react promptly.

Thank you again

The authors

Reviewer 2 Report

important references are missing and should be added:

-Ximeng Liu, Robert H. Deng, Kim-Kwang Raymond Choo, Yang Yang*, HweeHwa Pang: Privacy-Preserving Outsourced Calculation Toolkit in the Cloud, IEEE Transactions on Dependable and Secure Computing, 2018

Yang Yang, Xianghan Zheng, Wenzhong Guo, Ximeng Liu, Victor Chang .Privacy-preserving smart IoT-based healthcare big data storage and self-adaptive access control system.Information Sciences.Volume 479, April 2019, Pages 567-592

Chen X, Li A, Zeng X, et al. Runtime model based approach to IoT application development[J]. Frontiers of Computer Science, 2015, 9(4): 540-553.

Q.Jing,et al:Security of the Internet of Things: perspectives and challenges.

Wireless Networks 20 (8), 2481-2501, 2014

Chao Lin, et al:BSeIn: A blockchain-based secure mutual authentication with fine-grained access control system for industry 4.0. J. Network and Computer Applications 116: 42-52 (2018)

R.Deng,et al:False data injection on state estimation in power systems—Attacks, impacts, and defense: A survey. IEEE Transactions on Industrial Informatics 13 (2), 411-423,2016

Author Response

 Dear Valued Reviewer,

Thank you for your time to review our article. Without your noble work, our article could not move forward to this stage. We really appreciate your excellent work.

Hereby, we would like to transfer our answers to your questions. Once again, with your questions, the article has been updated accordingly. Should you find any other recommendations or suggestions, please let us know, we will react promptly.

Thank you again

The authors

Reviewer 3 Report

The paper proposed a cybersecurity solution based in the intrusion detection system. The proposed system is designed to detect  malicious activities targeting the Distributed Network Protocol. The DNP3 protocol is wildly used in the smart grid network and designing a cybersecurity which take in consideration the DNP3 characterisation and design is very interested research topic.

The proposed system is well explained in details. And a full details about  the tools used in the experiments were provided. One issue the authors need to consider is to justify why they picked 50% for the feature selection threshold.

Author Response

Dear Valued Reviewer,

Thank you for your time to review our article. Without your noble work, our article could not move forward to this stage. We really appreciate your excellent work.

Hereby, we would like to transfer our answers to your questions. Once again, with your questions, the article has been updated accordingly. Should you find any other recommendations or suggestion, please let us know, we will react promptly

Thank you again

The authors

Round 2

Reviewer 1 Report

The authors have made related revision according to my comments.